# How Good Is a Tactical-Grade GNSS + INS (MEMS and FOG) in a 20-m Bathymetric Survey?

**DOI:** 10.3390/s23020754

**Published:** 2023-01-09

**Authors:** Johnson O. Oguntuase, Anand Hiroji, Peter Komolafe

**Affiliations:** Division of Marine Science, School of Ocean Science and Engineering, University of Southern Mississippi, Stennis Space Center, MS 39529, USA

**Keywords:** tactical-grade MEMS, tactical-grade FOG, GNSS + INS, bathymetric survey, ellipsoidally referenced survey (ERS), multibeam echo sounder (MBES)

## Abstract

This paper examines how tactical-grade Inertial Navigation Systems (INS), aided by Global Navigation Satellite System (GNSS) modules, vary from a survey-grade system in the bath-ymetric mapping in depths less than 20 m. The motivation stems from the advancements in sensor developments, measurement processing algorithms, and the proliferation of autonomous and uncrewed surface vehicles often seeking to use tactical-grade systems for high-quality bathymetric products. While the performance of survey-grade GNSS + INS is well-known to the hydrographic and marine science community, the performance and limitations of the tactical-grade micro-electro-mechanical system (MEMS) and tactical-grade fiber-optic-gyro (FOG) INS aided with GNSS require some study to answer the following questions: (1) How close or far is the tactical-grade GNSS + INS performance from the survey-grade systems? (2) For what survey order (IHO S-44 6th ed.) can a user deploy them? (3) Can we use them for navigation chart production? We attempt to answer these questions by deploying two tactical-grade GNSS + INS units (MEMS and FOG) and a survey-grade GNSS + INS on a survey boat. All systems collected data while operating a multibeam system with the lever-arm offsets accurately determined using a total station. The tactical-grade GNSS + INSs shared one pair of antennas for heading, while the survey-grade system used an independent antenna pair. We analyze the GNSS + INS results in sequence, examine the patch-test results, and the sensor-specific SBET-integrated bathymetric surfaces as metrics for determining the tactical-grade GNSS + INSs’ reliability. In addition, we evaluate the multibeam’s sounding uncertainties at different beam angles. The bathymetric surfaces using the tactical-grade navigation solutions are within 15 cm of the surface generated with the survey-grade solutions.

## 1. Introduction

Varied applications are increasingly taking advantage of the affordable and power-efficient MEMS Inertial Navigation Systems (INS) aided by Global Navigation Satellites (GNSS). Hydrographers, marine scientists, geospatial professionals, and researchers are not alone in the quest to use affordable sensors that meet their operational requirements. As hardware performance and algorithms improve [1], many now consider low-cost and tactical-grade sensors for varied applications, including hydrographic operations. “There is no universally agreed definition of low-, medium-, high-grade IMUs and inertial sensors [2]”. Generally, those classifications follow performance (accelerometer bias in run stability) and cost. Here, we define low-grade GNSS + INS sensors as those with accelerometer bias instability worse than 3 mg (0.03 m s^−2^), gyro bias instability worse than 50°/h, and cost below USD 2000. We classify medium- or tactical-grade as those having accelerometer bias between 0.1 and 10 mg, gyro bias between 0.1°/h and 10°/h, and a price range between USD 7000 and USD 30,000. Finally, we classify sensors with bias in-run stability better than 0.1°/h, costing between USD 50,000 and USD 100,000 or higher as survey-grade.

Ascertaining a system’s performance and product reliability within specified minimum standards requires an a priori estimation of the systems’ uncertainty. In the context of marine applications and especially from the hydrographer’s perspective, fulfilling an a priori estimation for new low-cost and tactical-grade sensors demands evaluating the propagated uncertainties resulting from the bathymetric and navigation systems.

Studies exist on affordable MEMS, or tactical-grade GNSS + INS, for vehicle navigation in the urban environment, and examples of such studies include [3,4,5]; they discuss the performance to expect in GNSS-denied environments. However, very little is known about low-cost or tactical-grade GNSS + INS performances for hydrographic survey applications, especially for users and researchers seeking to take advantage of affordable sensors. In this study, we address the application of affordable MEMS and FOG INS, emphasizing their performance in multibeam sound and ranging (SONAR) survey procedures in shallow waters at sea, the limitations of the sensors, data acquisition, and processing challenges.

GNSS + INS sensors are best suited for terrestrial, and above-sea navigation rather than sub-sea navigation since GNSS signal propagation in the atmosphere experiences tolerable delays compared to in-water propagation, which attenuates quickly. That advantage favors GNSS + INS applications for most terrestrial and airborne applications, especially as uncrewed surface vehicles (USVs) proliferate. Today, two technologies dominate the INS market: MEMS and FOG. Scientists and informed users know that stand-alone MEMS INS performance is poor compared to FOG INS [6], but when integrated with GNSS sensors, its performance can approach FOG INS’s, provided that GNSS availability and integrity are continuous.

Multibeam echo sounder (MBES) bathymetric survey aboard a vessel, including USVs for ocean mapping, requires attitude, heading, position, and time determination from a GNSS + INS system. In addition, the positioning and navigation system integration must precisely time-sync the acoustics pulses arriving at the MBES receive array. One challenge with seafloor position determination using an MBES is the uncertainty in the sound pulse localization at the seafloor, especially in the outer beams, when operating the SONAR at a typical swath (< ±70 degrees). Therefore, accurate roll measurements and SONAR beam stabilization across track become critical to minimizing seafloor-sounding uncertainties. The direct comparison of low-cost and tactical-grade MEMS INS sensors to a survey-grade system is insufficient in evaluating systems’ reliability without determining the impact of roll measurements in the outer beams of the SONAR system. Hence, will low-cost or tactical-grade sensors meet roll requirements for the MBES surveys in shallow waters (<25 m)?

Putting that question in context demands examining low-cost GNSS + INS attitude resolution capability depending on MBES beamwidth’s resolution, beam-to-beam separation angle, and positioning strategy for aiding an INS. A survey of previous studies on affordable tactical-grade GNSS + INS shows different positioning and attitude performances depending on the positioning and integration strategies. GNSS positioning strategies for aiding INS include precise point positioning (PPP), PPP with ambiguity resolution (PPP-AR), and realtime kinematics (RTK). The latter is akin to the post-processed kinematic (PPK) in that they require carrier phase and pseudo-range double-differencing, except that some authors prefer to differentiate the RTK from the PPK strategy when the solutions are real-time as opposed to post-mission. In addition to the positioning strategies, different GNSS + INS integration schemes also offer different performances. Again, the overall performance varies with the implemented coupling scheme, whether loose, tight, deep, or ultra-tight.

While we do not attempt to describe those coupling schemes in detail, it is worth mentioning that the loose coupling scheme is an architecture that utilizes GNSS position and velocity solutions as inputs in the integration algorithm. The tight coupling scheme uses the GNSS observables (pseudo-range, carrier phase) and range rates rather than the final solutions (positions and velocities) as inputs in the integration algorithm. Deep coupling refers to the integration strategy in the hardware tracking domain, while the ultra-tight strategy combines the tracking domain and the range-domain integration schemes. Groves [2] describes in detail those coupling schemes and their algorithms.

Gao et al. [7] present multi-GNSS PPP and INS integration accounting for INS hardware errors, the intersystem, and inter-frequency biases common to the multi-GNSS PPP modeling and show that 3D-positioning RMS is within 10 cm for a low-cost MEMS and tactical-grade GNSS + INS hardware. In their work, velocity and attitude improvements are insignificant regardless of whether the PPP-INS integration is GPS-only or multi-GNSS. In addition, the 3D position, velocity, and attitude drifts during a 60-s GNSS outage are about 37 m, 1 m/s, and 0.2 degrees.

Previous studies showed promising results with the low-cost single-frequency RTK strategies [8,9,10,11,12,13]. Although those studies mainly evaluate RTK with low-cost single-frequency receivers, they provide insight into positioning performances one should expect when integrated with a low-cost INS using strategies identical to those discussed; in some cases, decimeter level and, in rare cases, sub-decimeter [9], depending on algorithm implementation.

Oguntuase et al. [14,15] show that using low-cost multi-frequency, multi-GNSS receivers on moving vehicles, PPK solutions with better than 20 cm uncertainties are possible. While an INS aided with low-cost single-frequency GNSS sensors may be desirable for other applications, they may not necessarily provide cost-effective solutions for hydrographers compared to aided INS using dual- or multi-frequency low-cost GNSS sensors. The ionospheric error mitigation, easily achievable with a dual-frequency when RTK is limited, makes it a better option than single-frequency-aided INS. In addition, intersystem bias (ISB) is one of the challenges researchers have noted with using mixed receivers for RTK/PPK. While ISBs may be stable over time and have a near-zero bias for identical survey-grade receivers, they are non-zero and unstable for some low-cost receivers [13]. In addition, Cécile et al. [13] chronicled previous studies affirm that ISBs are sensitive to firmware updates.

Similarly, the INS-GNSS integration scheme impacts the system’s expected performance. For example, Cécile et al. [13] present an INS-aided ambiguity resolution and tight-coupling algorithm for combining GPS and BeiDuo measurements with linear and angular accelerations from an INS, claiming rapid ambiguity fixing with their algorithm during a land-based test. Their results indicate that relative positioning uncertainty is between 4 and 6 cm. Though studies exist on low-cost GNSS + INS algorithms, filters, and performances using different GNSS positioning strategies, very little is known about how motion accuracies from tactical-grade GNSS + INS (MEMS and FOG) impact bathymetric surveys with MBES. Therefore, we address the following questions:How close or far is the tactical-grade GNSS + INS positioning performance from the survey-grade systems on a hydrographic survey platform in calm weather?Will the tactical-grade sensor meet the International Hydrographic Organization’s (IHO) Standard for Hydrographic Survey (S-44 6th ed.) in shallow water?Can we use those sensor grades for navigation chart production?

## 2. Methods

Since Hare et al. [16] comprehensively discussed MBES georeferencing using GNSS + INS solutions and the mathematical formulations of error propagations from all possible parameters involved in bathymetric mapping, we focus mainly on the performances achievable with those tactical-grade sensors integrated with an MBES as the novelty of our study. However, analyzing those sensors independently without integrating an MBES may not be sufficient to justify those sensors’ empirical performance for bathymetric surveys.

### 2.1. MBES Georeferencing Overview

The georeferencing model required to compute multibeam footprints’ geodetic coordinates involves combining the GNSS + INS navigation solutions with the MBES raw datagram. As shown in the conceptual representation in Figure 1, three frames are involved: the GNSS + INS frame ($F_{i})$, defined by the right-handed system with axes $X_{i},\;Y_{i},\;Z_{i}$ (positive down); vessel frame $F_{v}:Y_{v},\;X_{v},\;Z_{v}$(positive up); and transducer frame $F_{t}:X_{t},\;Y_{t},\;Z_{t}$ (positive down); where $X_{i},\;Y_{v},\;\mathrm{and}\;X_{t}$, are positive in the bow direction, and $O_{i},\;O_{v},\;\mathrm{and}\;O_{t}$, are the frames’ origins or sensing centers.

The typical georeferencing procedure, as discussed in [16] (using slightly different notations here), is as follows:

Compute beams footprints’ coordinates ($\vec{P}$) in the transducer frame, $F_{t}$, using the respective acoustic steering angle (*θ*) and measured range ($r_{m}$) from two-way travel time in Equation (1):(1)$${\vec{P}}_{F_{t}}={\left[0,r_{m}\sin\theta ,r_{m}\cos\theta \right]}^{T}{}_{F_{t}},$$Compute the coordinates of the vessel reference point $O_{v}$ from translation vector ${\vec{O}}_{iv}$, and determine the vessel’s attitude ${\left(\alpha ,\;P,\;R\right)}_{F_{v}}$ from $F_{i}$; where α, P, R, are heading with respect to north, pitch, and roll (“angles measured in a rotated coordinate frame as defined by the Tate–Bryant convention [16]”)Transform the coordinates in $F_{t}$ to $F_{v}$ using Equation (2);
(2)$${\vec{P}}_{F_{v}}=R_{t}^{v}\left(\alpha ,\;P,\;R\right)\;{\vec{P}}_{F_{t}}+{\vec{O}}_{tv},$$
where $R_{t}^{v}$ is the rotation matrix.

An alternative route is to transform coordinates in $F_{t}$ to $F_{i}$ and translate the results to $F_{v}$ via ${\vec{O}}_{iv};$

4.Compute refraction corrections, $n\left(SSP,\;\theta ,\;r_{m}\right)$, relative to measured ranges, as a function of the sound speed profile (*SSP*) and acoustic steering angles using the geometric raytracing algorithm discussed in [17] (pp. 47–52);5.Apply refraction corrections to sounding positions: ${\vec{P}}_{F_{v}}$ + $n\left(SSP,\;\theta ,\;r_{m}\right);$6.Account for small angular misalignments, $M_{t}^{i}$, between the transducer and the INS frame; the so-called patch test/ boresight calibration offset: $M_{t}^{i}\;\left(d\alpha ,\;dP,\;dR\right);$7.In Equation (3), calculate the final multibeam-sounding coordinates (${\vec{P}}_{c}$) in the vessel frame, $F_{v}$, with boresight calibration, refraction corrections, and the separation model (*SEP*) applied, such that the final depths are reduced to a chart datum:(3)$${\vec{P}}_{c}={\vec{P}}_{F_{v}}+M_{t}^{i}\;\left(d\alpha ,\;dP,\;dR\right)+n\left(SSP,\;\theta ,\;r_{m}\right)-SEP_{ellipsoid-chart}.$$

To determine the performance of two tactical-grade GNSS + INS units in MBES footprints’ georeferencing, we will compare their GNSS + INS formal errors (position and attitude), roll, pitch, and ellipsoidal heights, the patch-test values from $M_{t}^{i}\;\left(d\alpha ,\;dP,\;dR\right)$, and the map surfaces from ${\vec{P}}_{c}$, relative to corresponding parameters from a survey-grade unit.

### 2.2. Experiment Design

All the INS simultaneously acquired data at Port of Gulfport, Mississippi, in the USA (shown in Figure 2). The experiment involved a pole-mounted Norbit iWBMS 12004 installed on a survey boat (RV LeMoyne), a survey-grade (Applanix’s WaveMaster II), and two tactical-grade INS: SBG Ellipse-D (MEMS-based) and KVH FOG 3D (MEMS-based accelerometers and FOG-based gyros). All units were mounted on an aluminum plate to ensure similar mounting angles, heave, pitch, and roll motions between systems. In addition, the tactical-grade sensors shared common primary and secondary antennas, independent of the antennas utilized for the survey-grade system. The simultaneous data acquisition involved a set of 6 lines at the patch test location in 42 different passes (making a total of 42 lines) at an average speed of 2.7 knots, and each line was approximately 250 m. Since bathymetric mapping on a hydrographic survey platform rarely occurs at speeds beyond 7 knots, our experiment design focused primarily on the best practices for typical hydrographic surveys. For instance, we did not examine GNSS + INS performance at turns since it is best to avoid multibeam data collection in such instances to minimize data cleaning time resulting from sparse seafloor coverage and sounding geolocation biases due to rapidly changing sounding in sonification angles and induced heave effects. Likewise, we did not examine performance in GNSS-denied environments since a typical nearshore survey occurs in open skies.

Table 1 shows the GNSS + INS systems and the paired antennas. We intentionally omitted their prices since we aim to investigate GNSS + INS in the low-price spectrum relative to a high-end system and do not intend to label one system as inferior or superior. Table 2 lists the group /binary/packet messages required to post-process the navigation dataset successfully for each GNSS + INS unit in their native software. Finally, Table 3 highlights the INSs’ accelerometers, gyroscopes, magnetometers, and pressure specifications [18,19,20,21,22,23], detailing the manufacturers’ claims about velocities, heading, attitude accuracies, et cetera. Following our previous definition of a medium- or tactical-grade system (0.1 < accelerometer bias < 10 mg; 0.1°/h < gyro bias < 10°/h), Ellipse-D’s accelerometer is better than the definition range for accelerometer bias, but its gyro bias is within range. FOG-3D’s biases are near the lower limits of the defined intervals.

With the aluminum plate containing all the INS installed on the vessel’s roof, we used total station techniques to determine INS sensing centers relative to the vessel’s reference origin, the antenna lever-arm offsets, and INS mounting angles relative to the vessel frame. The total station’s angular measurement precision is 0.5 s. Multiple total station observations from different stations allow for standard deviation estimation. Other small measurements, such as offsets from the IMU body to the sensing center, were measured precisely from step files using AutoDesk Fusion360 and comparing those measurements to sensors’ diagrams. As a result, those lever-arm offsets have less than 1 cm uncertainties. The data acquisition rate followed the manufacturers’ recommendations. That is particularly important to avoid data packet gaps, which would degrade the quality of the navigation solutions. We ensured the selection of appropriate baud rates—the highest baud rate possible on the laptop—for systems using serial com ports for data acquisition. In addition, we turned off data packets not required for successful post-processing to ensure that throughput does not overwhelm the selected baud rate. It is essential to mention that data logging and communications with the tactical-grade systems are via USB and serial ports by default, which poses a challenge in deciding throughputs for the relevant data packets and rates required to successfully post-process the GNSS + INS data in the respective native software. Data acquisition trials at 50-Hz and 200-Hz via serial port created data dropouts and gaps, precluding a continuous navigation solution. Thus, the 921,600 bauds work well with data packets and rates in Table 2 without data gaps.

We adopted the ellipsoidally referenced surveys (ERS) strategy for soundings reductions [24,25]. Since all the GNSS + INS systems experienced similar motion on the vessel, in theory, heave and tides effects on footprint geolocations’ uncertainties are insignificant compared to individual systems’ positioning accuracy and their ability to compensate for MBES’ roll and pitch motions since the GNSS + INS rate sufficiently captured those effects. Given that our goal is to investigate the two tactical-grade systems for MBES acquisition in a simultaneous experiment involving a survey grade system, we hypothesize that the smooth-best estimates of trajectories (SBET) from each system integrated with the MBES data in post-processing should reflect the GNSS + INS performances relative to the survey-grade system. Moreover, bathymetric processing packages usually provide an option to overwrite real-time position and motion records contained in the MBES datagram with the SBET records. Overwriting those records with tactical-grade SBET will yield different soundings geolocations based on system-specific positioning and motion performance. Finally, it suffices to mention that SBET is an industry-standard format containing post-processed navigation solutions (positions, attitude, and time tags) from a GNSS + INS system. Following the espoused hypothesis, the iWBMS’ factory-integrated INS (Applanix WaveMaster II; henceforth dubbed POSMV) streamed positions, time tags, and motions, enabling real-time soundings geolocations and beam stabilization. As mentioned earlier, the expectation is that the post-processed results from individual systems will overwrite the real-time navigation records.

### 2.3. Processing and Analysis Strategy

Figure 3 presents the data processing and analysis overview. All GNSS + INS datasets were post-processed to generate ASCII and binary files (SBET). The bathymetric data processing involved sound speed profile ingestion for raytracing and sounding geolocations with high-quality navigation results from the different GNSS + INSs. We used the manufacturer’s native software for respective sensors’ GNSS + INS data post-processing to generate ASCII and SBET files, using the post-processed kinematic positioning (PPK) strategy. For example, the smart-base processing engine in Applanix POSPAC MMS (version 8.7) for the Applanix POSMV, SBG Qinertia Pro’s (version 3.1.7593) tight coupling PPK for SBG Ellipse-D, and Advanced Navigation Kinematica’s single-base PPK for KVH’s FOG 3D. Notably, Advanced Navigation did not provide the version number of their online Kinamatica software, which uses the single-base processing strategy. In addition, Kinamatica does not implement the virtual reference station (VRS) or network PPK strategy as the other software packages. Therefore, our processing utilized the single-base processing strategy relying on a continuously operated local station installed on the University of Southern Mississippi’s Marine Research Centre building (about 2 km away). POSPAC MMS smart-base and Qinertia tight coupling processing engines are network processing strategies based on the technique described in [26,27,28,29,30,31].

We used CARIS HIPS and SIPS (version 11.4.11) for multibeam data processing. The navigation solutions in the ASCII format easily allow systems’ results comparisons. However, the SBET files, whose contents are similar to ASCII’s, are preferable when ingesting post-processed navigation solutions in CARIS HIPS and SIPS, evading the data parser step in the ASCII file ingestion case. We used the ERS with a single static offset for bathymetric soundings reductions to chart datum. That sounding reduction technique relies on GNSS vertical referencing, which captures heave motion provided the GNSS update rate is high enough (>10 Hz). Since all GNSS + INS systems provide navigation solutions at rates ≥50 Hz, they sufficiently capture heave signals. Hence, the heave results from the inertial sensors (GNSS-independent) for the three navigation systems are not discussed here.

We anticipate processing strategy implementation differences between varying processing engines. Therefore, it is conceivable that their navigation results will vary slightly. In addition, GNSS + INS hardware differences and their uncertainties will also propagate into sounding footprints’ geolocations. That is, the situation in practice where different users are expected to provide comparable bathymetric products, even though they have deployed different systems and software suites from various manufacturers. As mentioned earlier, this study allows us to examine what those bathymetric differences might be relative to soundings with a survey-grade navigation system operating simultaneously with the tactical-grade systems.

## 3. Results

The analysis involves the following:Formal errors (uncertainties) from software’s GNSS + INS stochastic and noise models;Direct comparisons of roll, pitch, and heights between systems;CARIS’s boresight calibration results (patch test) from a multibeam dataset georeferenced with system-specific SBET;Bathymetric surface and beam analysis.

### 3.1. Formal Errors

Though the term “uncertainty” is preferable over “error”, here, we refer to the reported standard deviations from the GNSS + INS post-processing software as formal errors to differentiate them from the total propagated uncertainty (TPU) estimations for MBES soundings and the standard deviations associated with the bore-sight (patch test) calibration process. It is worth noting that the soundings’ TPUs mentioned here refer to the combined uncertainties in MBES rangings, sound velocities, lever-arms estimations, GNSS + INS solutions, bore-sight calibration, and vertical datum separation model. In addition, the Combined Uncertainty and Bathymetry Estimator (CUBE) [32], available in most commercial software, requires formal errors (SBET RMS files) in determining soundings’ propagated uncertainties. Hence, soundings’ TPU are directly dependent on software formal error output and do not necessarily imply soundings’ quality superiority when compared with outputs from different systems. One way to determine navigation quality superiority in a field experiment is by incorporating an independent validation process as an “expert witness” to the systems’ navigation results. However, that can be challenging as all systems’ time and measurement processes must be synchronized accurately.

Since formal errors represent dispersions for a measured parameter by a particular system and are related to the Kalman filter stochastics and noise models determined by the instrument manufacturer and software authors, we suspect those formal errors are not perfect metrics when comparing different systems from different manufacturers. However, we present parameter-specific and system-specific average formal errors for instantaneous navigation results in Figure 4 as a snapshot and metric for system performance to illustrate our point.

The average formal errors are calculated using Equation (4):(4)$$sd_{average}=\frac{\sqrt{\sum_{i=1}^{n}\sigma_{i}^{2}}}{n}\;,$$
where *i* is the *i*th record, and *n* is the total number of records.

It is evident from Figure 4 that POSMV’s formal errors are best across all parameters except for its pitch and heading, which fall behind FOG-3D’s formal errors, though the manufacturer’s specification suggests that FOG-3D’s heading accuracy is better than POSMV’s (see Table 3). In contrast, FOG-3D’s formal errors are almost twice Ellipse-D’s in the north, east, and height components, though FOG-3D’s attitude formal errors are comparable to POSMV’s. Following similar performances in their (FOG-3D and POSMV) attitudes, one would expect comparable formal errors in their north, east, and height components, which drives home the earlier point that formal errors may not be perfect metrics when comparing two systems. Those formal errors would vary between systems for reasons, including, but not limited to, uncertainties in the accelerometer bias, gyroscope bias, GNSS tracking and measurement processes, Kalman filter implementation processes, system noise, and GNSS + INS processing strategy implementation and stochastics based on the estimated system’s performance. Furthermore, the formal errors from the system-specific SBET RMS files may not provide the true formal errors for CUBES’ hypothesis in determining the propagated soundings’ uncertainties [33].

### 3.2. Attitude Results

Figure 5 and Figure 6 present direct systems’ comparisons of the roll and pitch time series. A zoomed-in view of Figure 5, as shown in the lower panel snippets, reveals that POSMV and Ellipse-D roll values are almost identical (green line overlay broken blue line), while FOG-3D values are at almost a constant offset from them. The mean roll offset between POSMV and Ellipse-D is 0.01 degrees, and the corresponding 95% ordered statistics are 0.01 degrees. FOG-3D’s mean roll offset from POSMV is 0.19 degrees, and the corresponding 95% ordered statistics are 0.51 degrees.

Figure 6 shows that the pitch offsets between POSMV and Ellipse-D are consistent throughout the time series but inconsistent between POSMV and FOG-3D. For example, the mean offsets and 95% ordered statistics for those system pairs are (0.23, 0.24) degrees and (0.09, 0.23) degrees, respectively. Overall, the FOG-3D’s roll and pitch offsets from POSMV are larger than Ellipse-D’s.

Table 4 summarizes the offsets between the tactical-grade systems and the survey-grade system. Though the autocalibration accounts for the offsets in MBES processing, here, we briefly show as an example how roll offsets typically propagate into depth determination as a function of roll error (in this case, offsets) up to 0.51 degrees, from Equation (5) [26] (p. 365).
(5)$$\delta z_{\theta }=\delta \theta *H*tan\theta \;,$$
where *H* is depth, *θ* is half the swath angle, and *δθ* is the angle measurement error (offset). For example, for an MBES using a 130-degree swath at an 11 m depth, a roll offset up to 0.51° is equivalent to a 10 and 21 cm depth error. In other words, the systematic errors in depth as a function of roll offsets between the systems under investigation vary between 10 and 21 cm. Those offsets are accounted for in boresight calibration, as discussed in Section 3.4. Since this study compares how close or far the tactical-grade systems are from a survey-grade system, the values discussed here are relative offsets to the survey-grade system and do not in any way imply absolute uncertainty for any given system.

### 3.3. Ellipsoidal Height Time Series

Given that the vertical component is the most critical in ellipsoidally referenced bathymetric mapping, snippets in the lower panels of Figure 7 emphasize spike and drift locations in the upper panel. We expect the solution outputs would vary slightly because all systems did not apply the mounting angle offsets. POSMV and Ellipse-D have similar height results, which vary with FOG-3D’s results by a magnitude up to 8 cm. As seen in Figure 7, Ellipse-D and FOG-3D’s ellipsoidal heights are similar (95% ordered statistics are 2 cm) but differ slightly from POSMV’s results. The mean height offset between POSMV and Ellipse-D is 4.6 cm, and the 95% ordered statistics are 8.1 cm. That statistic is similar to the mean offset between POSMV and FOG-3D and the corresponding 95% ordered statistics.

Designating the survey-grade GNSS + INS results as the “truth” (reference solution), which in the theory of error is unknown. For comparison, we assume that those offsets are the uncertainties from the reference solution. Again, we emphasize that the GNSS + INS results did not account for mounting angle offsets but are compensated for in boresight calibration during the bathymetric processing. Following that assumption, the worst depth uncertainty variation between the systems under review, combining their worst attitude- and ellipsoidal height-dependent errors ($\sqrt{21^{2}+8^{2}}$) in 11-m water depth is 22 cm. That value (22 cm) is the worst depth error relative to the survey grade system, and it does not imply the absolute overall uncertainty expected from any given system. However, we can infer that systematic error when using a tactical-grade GNSS + INS system without explicitly accounting for the mounting angles can reach 22 cm relative to a survey-grade system.

The spikes in GNSS heights, as seen in Figure 7**,** are often related to degraded measurement quality due to multipath, which typically result in cycle slips and, thus, float solutions and a drop in the number of satellites tracked. Such an isolated spike is noted in the FOG-3D time series more than in the other two systems. That may be related to the processing strategy of the different processing software suites. It suffices to reiterate that we processed the FOG-3D with Kinematica, an online processing suite, using a single base station. We are unaware whether the software implements network PPK, smart-base, or a virtual reference station processing strategy as typical with the Applanix POSPAC MMS and SBG’s Qinertia software suites. Moreover, Kinematica does not currently provide results in an SBET format but in a CSV format which we translated into the SBET files used in the bathymetry processing.

### 3.4. Boresight Calibration

Figure 8 shows sensor-specific roll, pitch, and heading autocalibration results and their respective one-sigma uncertainties (error bars in Figure 8). Compared to traditional calibration tools, the automatic boresight calibration [34,35] algorithm eliminates operators’ bias in determining patch-test calibration offsets stemming from the mounting angles between the vessel and sensors’ frames. In the INS setup, the vessel configuration survey did not account for the mounting angles; hence, they would appear in the calibration offsets. Since all INS sensors are mounted on the same plate, we expect similar calibration offsets, assuming the variability between the INS body reference frames is negligible. That is evident in the roll and pitch calibration offsets for all systems, as seen in Figure 8. The exception is the heading offset which is different for the three systems, with Ellipse-D having the lowest and FOG-3D having the highest offset and highest one-sigma uncertainty. One possible reason for Ellipse-D’s insignificant heading offset is the automatic lever-arm offset and dual antenna alignment estimations applied in the processing.

The calibration uncertainties suggest that the roll has the best calibration parameter, followed by the pitch, with the heading’ calibration being the least accurate. However, the pitch calibration values appear similar for the three systems, but their uncertainties are more widely apart than the roll calibration uncertainties.

### 3.5. Bathymetry and Beams Uncertainty

To further investigate the relative depth uncertainty for the systems under discussion, we generated and compared bathymetric surfaces using system-specific SBETs. It is again important to emphasize that the autocalibration accounted for roll, pitch, and heading misalignments. Here, we discuss system-specific bathymetry surfaces in the context of the IHO’s minimum bathymetry standard shown in Table 5 [36]. According to the sixth edition of IHO S-44, minimum bathymetric standards specify the maximum total allowable vertical uncertainty as $TVU_{max}\left(d\right)=\sqrt{a^{2}+{\left(b*\;d\right)}^{2}}$, where *a* is a depth-independent parameter, *b* is a depth-dependent coefficient, and *d* is depth. Table 5 shows the allowable TVUs for the bathymetry range at the experiment location. The TVUs at 5 and 11 m depths for Exclusive, Special, and Order-1 surveys are (15, 17 cm), (25, 26 cm), and (50, 52 cm).

Figure 9 shows the system-specific bathymetric surfaces (0.25 m resolution) using the CUBE algorithm. The upper panel, ordered left to right, indicates identical surfaces georeferenced with SBET and RMS files from POSMV, Ellipse-D, and FOG-3D systems and the corresponding difference maps between the survey-grade and the tactical-grade systems. The color ramp, from red to green, and blue regions indicate shallow (about 5 m) to deep (11 m) bathymetry. The physical inspection of the difference surface maps (center and right panels) does not reveal spatial differences, but the uncertainty surfaces (Figure 10) show that differences exist. Portions of the surfaces meeting the IHO’s minimum bathymetry standard for Exclusive and Special Order are in green (0–0.15 m) and blue (0.15–0.25 m), respectively. Notably, the tactical-grade systems did not quite meet the Exclusive Order standard in all locations, especially at the outer edges of the deep area, indicating inferior performance in the outer beams of the MBES at those locations compared to the shallow area. Thus, the uncertainty in the roll angles for the tactical-grade systems increases with the sounding range. TVU surfaces show that the worst uncertainties (15–25 cm) in the outer edge of the deep area (11 m) agree with the earlier uncertainty estimations as a function of combined uncertainties in roll-pitch + GNSS-heights relative to the survey-grade system. It is essential to again, highlight that CUBE’s TVU estimation relies on system-specific formal errors (SBET RMS). Those formal errors stem from the manufacturer’s Kalman filers and the software stochastic in qualifying navigation solution confidence. Again, we suspect that improvements in those formal errors will directly impact CUBE’s TVU estimations. Hence, an external validation process is desirable in quantifying systems’ performance at a higher degree of confidence.

We further analyze the surfaces and percentages of soundings meeting Special and Order-1 requirements as a function of beam angles. Finally, the cumulative distribution function (Figure 11, left panel) compares the tactical-grade bathymetric surfaces to that generated from the Applanix system. The analysis reveals that surfaces generated from the Ellipse-D’s SBET are closer to Appanix’s than the FOG-3D’s surface. That agrees with the roll performances discussed earlier. The cumulative distribution function shows that more than 95% of Ellipse-D’s bathymetric surfaces agree with POSMV’s surfaces within ±10 cm. Similar statistics for FOG-3D show agreement within ±15 cm. That, again, is a reflection of the roll performances discussed earlier.

## 4. Discussion

Table 6 summarizes the main findings from the results and analyses in the previous section: boresight calibration results, the ellipsoidal height differences relative to the survey-grade system, surface differences from tactical-grades’ bathymetric surfaces relative to the POSMV’s surface, and the beam analysis for IHO’s Special and Order-1 requirements.

In the context of a hydrographic survey in shallow waters (<25 m) where the tolerance limit for the TVU is 50 cm, we note that tactical-grade systems are suitable, provided the navigation dataset has near zero-data gaps. That is particularly true from the analysis presented in Figure 11 (also summarized in Table 6), which shows that Ellipse-D’s and FOG-3D’s sounding footprints, within a ± 65° swath, pass the IHO’s Order-1 requirements. The percentage of soundings for beams between the ± 65° swath, which pass that requirement, are between 98.6% and 99.8%. In contrast, the survey-grade system (POSMV) has soundings passing the Order-1 requirement almost 100% for all beams. Professionals expect that level of performance for a high-quality GNSS + INS unit. In contrast, tactical-grade systems infrequently meet the Special Order requirements, according to the performance analysis shown in Figure 11 (top right panel). For example, 80–95% of FOG 3D’s beams between the ± 65° swath passed that requirement, while Ellipse-D’s case is 86–98%, suggesting their inadequacies for Special Order or Exclusive Order surveys even at 25 m water depth.

As mentioned earlier, since the CUBE algorithm relies on formal errors (SBET RMS) in estimating the uncertainties for the individual footprints or beams, we expect that a system’s performance and ability to meet higher order requirements would improve if the corresponding formal errors improve. It is essential to again mention that one would expect formal error variations between systems for various reasons, including, but not limited to, the following: system noise, bias calibration residuals, random noise, system quality, processing strategy implantation, Kalman filter stochastics, etcetera. Though the beam analysis in Figure 11 suggests that surveyors should restrict tactical-grade systems to non-exclusive and non-special orders, we note that the bathymetric surfaces differences relative to the survey grade system are ±10 cm (Ellipse-D) and ±15 cm (FOG-3D). According to Figure 9, Figure 10 and Figure 11 (left), more than 95% of Ellipse-D’s soundings are well within ±10 cm of the POSMV’s soundings. Similarly, more than 95% of FOG-3D’s soundings are within ±15 cm, suggesting that the localization performances of these systems approach that of the survey-grade systems.

In addition, the roll and pitch estimations are similar, except for negligible static offsets between systems due to the mounting angle not applied during the acquisition but applied in multibeam processing using the autocalibration technique. We noted those similarities in the roll and pitch calibration results. However, the heading values are dissimilar, which is understandable as the mounting angles were unaccounted for in real-time. Data gaps, logging rates, packet/group data configuration, and the throughput estimation to determine the proper baud rates when logging via a serial port impinge on the navigation solutions’ quality for almost any integrated systems, and the tactical-grade systems used here are no exception. However, those data gap effects and navigation degradation at turns are not discussed here as we focus primarily on the best results possible with tactical-grade systems and how close they are to the survey-grade results. In addition, surveyors typically avoid collecting multibeam data at turns to avoid the time implications required to clean noisy data.

## 5. Conclusions

We examined the performance of two tactical-grade systems relative to a survey grade in depths less than 20 m. The experiment used common antennas for the two tactical-grade systems independent of the survey-grade antennas, with all GNSS + INS units mounted on the same plate on a survey vessel. Integrating the post-processed SBET results with the multibeam dataset, we note that more than 99% of tactical-grades’ results conveniently meet IHO Order-1 bathymetric requirements, while 86% to 92% of the tactical-grades’ results also meet the Special Order requirements for beams within a ± 65° swath. Additionally, we used the survey-grade results as a reference and determined that the tactical-grade multibeam soundings are better than the ±15 cm of the survey-grade soundings 95% of the time. That level of accuracy is possible in the open sky and requires a near-zero data gap for the GNSS + INS dataset. However, whether the tactical-grade systems pass the IHO’s Special Order requirement is subject to system-specific formal errors upon which CUBE’s hypothesis relies in estimating TPU. Therefore, a validation technique independent of relative comparisons between GNSS + INS sensors may be necessary for future studies to justify using tactical-grade sensors for Special Order surveys and navigation chart products, especially where under-keel clearance is critical. In addition, future studies will attempt a direct integration of tactical-grade units with a multibeam system to access their real-time performances.

## Figures and Tables

**Figure 1 sensors-23-00754-f001:**
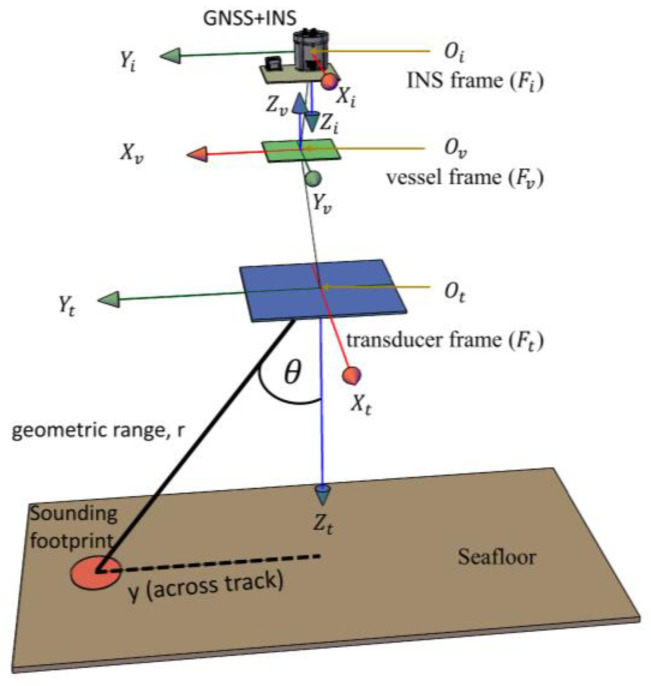
A conceptual overview of MBES footprints’ georeferencing using GNSS + INS mounted on a survey platform with the MBES fixed to the vessel body frame.

**Figure 2 sensors-23-00754-f002:**
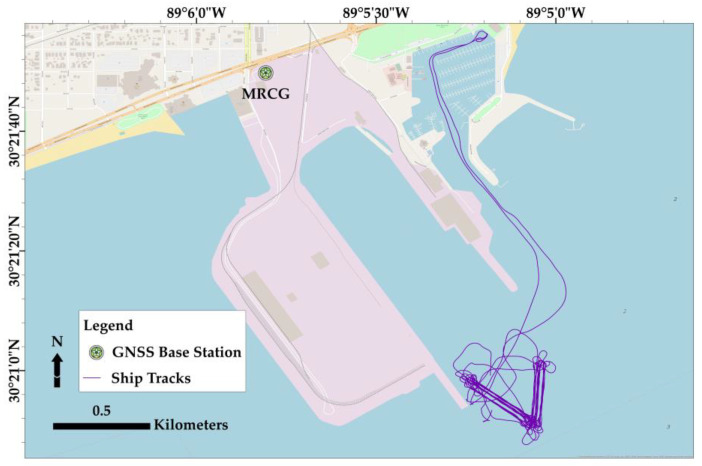
Survey track lines at the experiment location. The water depth is up to 11 m, and the swath width for Norbit iWBMS 12004 acquisition is 130 degrees.

**Figure 3 sensors-23-00754-f003:**
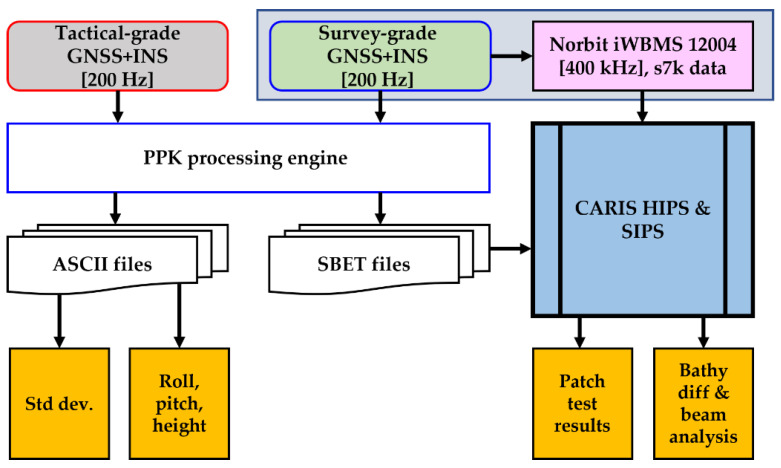
Data processing (PPK) and analysis overview—navigation results uncertainties; roll, pitch, elevations, patch test results, and bathymetric data comparison between systems offer insight into systems’ performances.

**Figure 4 sensors-23-00754-f004:**
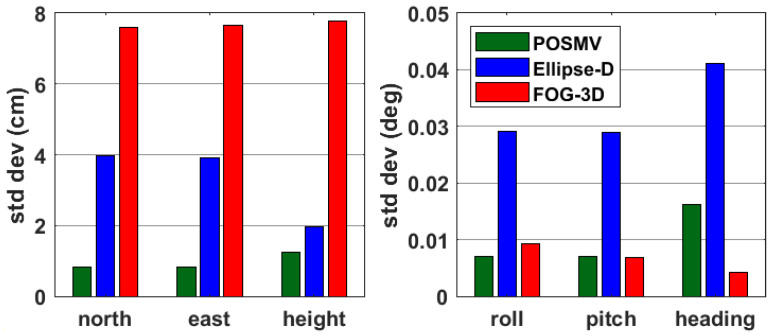
GNSS + INS average formal errors (standard deviations).

**Figure 5 sensors-23-00754-f005:**
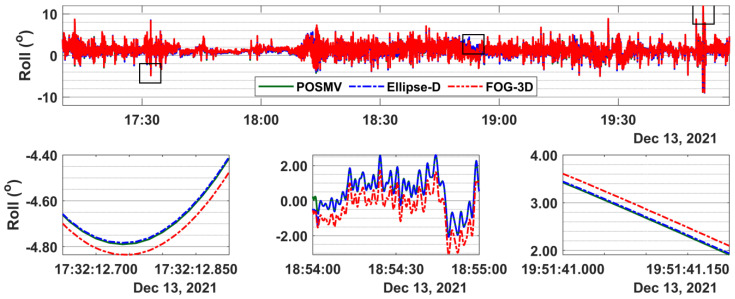
**Top** panel: roll time series for POSMV, Ellipse-D, and FOG-3D spanning about 3 h of observation. **Bottom** panel: shows snippets at the time series’ beginning, middle, and end. Roll values for POSMV and Ellipse-D are consistent throughout the dataset, with FOG 3D at a constant offset.

**Figure 6 sensors-23-00754-f006:**
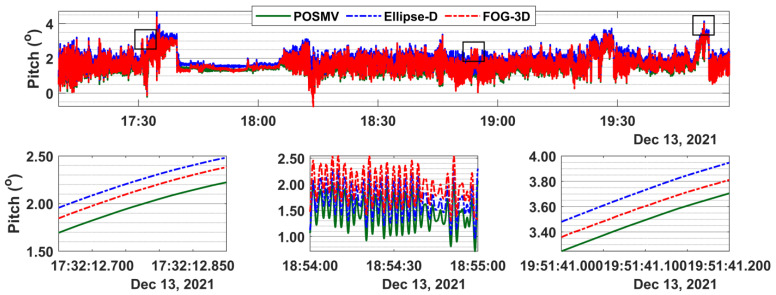
**Top** panel: pitch time series for POSMV, Ellipse-D, and FOG-3D. **Bottom** panel: shows 0.1-s, 30-s, and 0.1-s snippets at the time series’ beginning, middle, and end.

**Figure 7 sensors-23-00754-f007:**
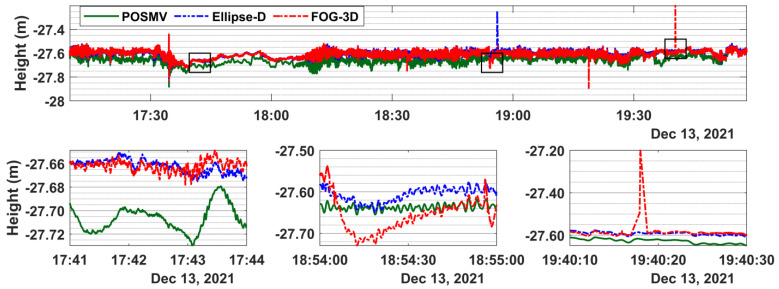
**Top** panel: ellipsoidal height time series for POSMV, Ellipse-D, and FOG-3D. **Bottom** panel: shows snippets at selected peaks. The mean offset between POSMV and the other systems is about 5 cm, and the 95% ordered statistics are about 8 cm.

**Figure 8 sensors-23-00754-f008:**
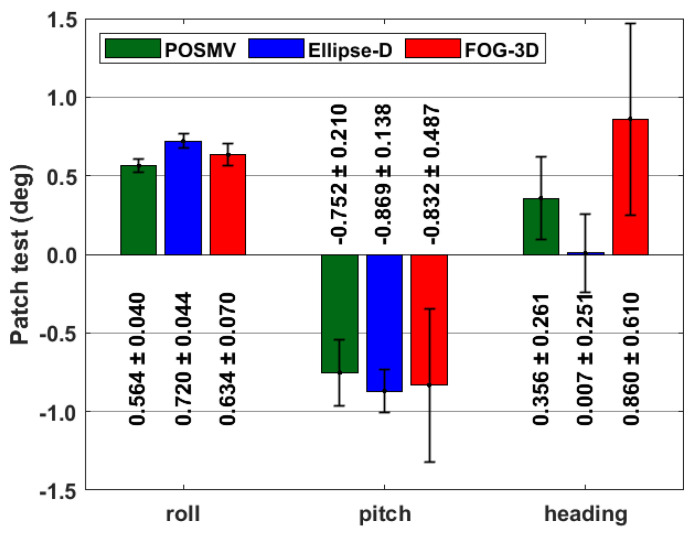
Roll, pitch, and heading calibration offsets and their respective 1-sigma uncertainties.

**Figure 9 sensors-23-00754-f009:**
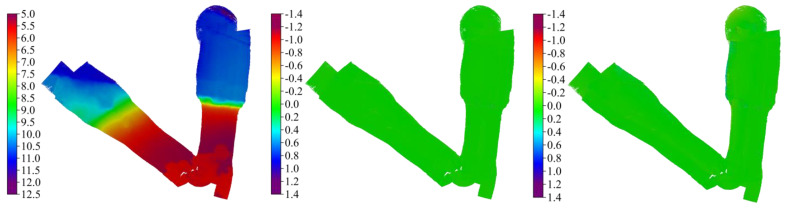
Ordered **left** to **right**: shows the bathymetric surfaces at 0.25-m resolution for POSMV (Wave-Master), the difference map for POSMV versus Ellipse-D, and the difference map for POSMV versus FOG 3-D.

**Figure 10 sensors-23-00754-f010:**
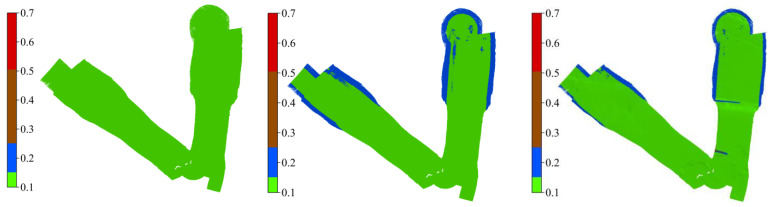
Ordered **left** to **right**: shows the TVU surfaces for POSMV, Ellipse-D, and FOG 3-D. The green areas imply TVUs between 0 and 15 cm, and the blue area implies TVUs between 15 and 25 cm.

**Figure 11 sensors-23-00754-f011:**
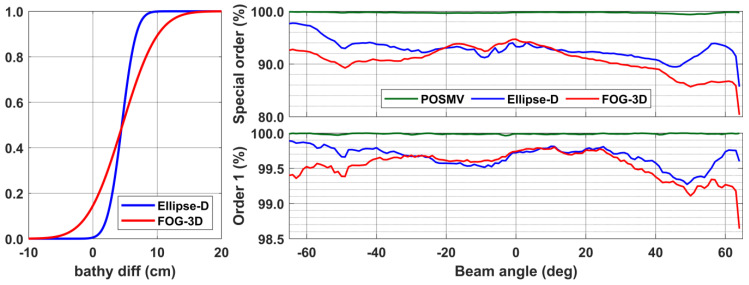
**Left** panel: Cumulative distribution function describing Ellipse-D and FOG-3D bathymetric surface difference relative to POSMV (Wave-Master). **Right** panels: percentages of soundings at passing Special (**top**) and Order-1 (**bottom**) at beams between the ± 65° swath.

**Table 1 sensors-23-00754-t001:** GNSS + INS sensors and antennas deployed during the experiment.

Manufacturer/GNSS + INS Brand	Grade	GNSS Model/ INS Data Rate	GNSS Antenna
Applanix/Wave Master II	Survey	*/ 200 Hz	Zephyr Model 2
SBG Systems/Ellipse-D (MEM-based accelerometers)	Tactical	Ublox ZED-F9P/ 50 Hz	Omnister G5Ant-53A4T1
KVH/FOG 3D (MEMS-based accelerometers and FOG-based gyros)	Tactical	Trimble MB-Two/ 200 Hz	Omnister G5Ant-53A4T1

Note: * unspecified data in documents accessible to the authors.

**Table 2 sensors-23-00754-t002:** Data packets and rates configuration for each GNSS + INS to enable post-processing.

Wave Master II	Ellipse-D		FOG-3D	
Group Number (Rate = 200 Hz)	Binary Log	Rate (Hz)	Packet ID	Rate (Hz)
1–5	System status, UTC	1	20 System state	200
9–11	EKF Euler	50	28 Raw sensors	200
99, 102	EKF navigation	50	29 Raw GNSS	20
110–114	Ship navigation	50	60 Raw satellite data	20
10,001, 10,007–10,009, 10,011–10,012	IMU Short, GPS1 velocity, GPS1 position, GPS1 true heading GPS1 raw data	On new demand		

**Table 3 sensors-23-00754-t003:** Systems’ specifications overview as provided in the manufacturer’s manual.

	Parameters	Wave Master II	Ellipse-D	FOG 3D
Accelerometer *	Input rate (g)	*	*	10 (max)
	In-run bias instability (μg|mg)	*	14|0.014	<50|0.05
	Random walk (μg/√Hz|mg/√Hz)	*	57|0.057	≤120|0.12
	Bandwidth (Hz)	*	390	≥200
Gyroscope	Input rate (°/s)	*	*	490 (max)
	In-run bias instability (°/h)	*	8	0.05
	Random walk (°/h/√Hz)	*	*	0.7
	Bandwidth (Hz)	200	133	≥440
Magnetometer	Range (G)	*	*	8
	Bias instability (mG)	*	1	*
	Random walk (μG/√Hz|mg/√Hz)	*	*	210|0.21
	Bandwidth (Hz)	*	22	110
Pressure	Range (KPa)	*	0.05–0.35	10 to 120
	Bias instability (Pa/yr)	*	<100	100
	Random walk (Pa/√Hz)	*	*	0.56
	Bandwidth (Hz)	*	100	50
GNSS aiding	Heading accuracy (1-m baseline, °)	0.015–0.03	0.4	0.01
	Velocity accuracy (m/s)	0.05	0.05	0.005
Navigation	Roll and pitch accuracy (°)	0.02–0.03	<0.1	0.01

Note G = Gauss; * unspecified data in documents accessible to the authors; mg = milli-g or μg = micro-g.

**Table 4 sensors-23-00754-t004:** Mean and 95% ordered statistics for attitude and elevation offsets from POSMV.

	Roll		Pitch		Heading		Elev	
	Mean	95%	Mean	95%	Mean	95%	Mean	95%
POSMV/Ellipse-D	0.005	0.012	0.233	0.247	0.694	0.430	0.048	0.082
POSMV/FOG-3D	0.194	0.505	0.094	0.234	0.087	0.159	0.046	0.084

**Table 5 sensors-23-00754-t005:** IHO’s minimum bathymetry standard for Exclusive, Special, and Order-1 hydrographic surveys.

	Exclusive	Special	Order-1
a	0.15	0.25	0.5
b	0.0075	0.0075	0.013
min depth (m)	5	5	5
max depth (m)	11	11	11
TVU (min {m})	0.15	0.25	0.50
TVU (max {m})	0.17	0.26	0.52

**Table 6 sensors-23-00754-t006:** Results and analysis summary.

	POSMV			Ellipse-D			FOG 3D		
Calibration (roll, pitch, heading {°})	0.564 ± 0.040	−0.752 ± 0.210	0.356 ± 0.261	0.720 ± 0.044	−0.869 ± 0.138	0.007 ± 0.251	0.634 ± 0.070	−0.832 ± 0.487	0.860 ± 0.610
Ellipsoidal heights with respect to POSMV	-			±11 cm			±10 cm		
Bathymetry with respect to POSMV	-			±14 cm			±18 cm		
Beams within 130-degree swaths passing IHO Special Order	99.4–99.9%			85.7–97.7%			80.3–94.7%		
Beams within 130-degree swaths passing IHO Order-1	99.6–100%			99.3–99.9%			98.6–99.8%		

Note: “-“ implies reference data for estimating tactical-grade units’ statistics.

## Data Availability

The data presented in this study are over 75 GB in size and not uploaded to an online repository but are available from the corresponding author on request.

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
