# Peer review of "How Good Is a Tactical-Grade GNSS + INS (MEMS and FOG) in a 20-m Bathymetric Survey?"

_sensors, 2023, doi:10.3390/s23020754_

Round 1

Reviewer 1 Report

This paper presents an important comparison between different grades of Aided Inertial Navigation Systems and the consequences of their uncertainty on the resulting seabed depth estimate. The work is important, the experiment is well structured, and the results are mostly clear. Some comments and clarifications follow. Each is tied to a line number from the PDF paper draft and section. 

Line 17: Does "survey order" refer to the IHO S44 v6 order levels in Table 1 of S44? You should specify what survey order means. 

32: Why is Inertial capitalized but not navigation systems? 

33: What are "allied professionals"?

35: You talk about "tactical-grade" and "survey -grade" aided INS, but what d these classifications mean? What makes a system tactical grade vs survey grade? This should be defined someplace. 

56: "Scientists and informed users know that stand-alone MEMS INS performance is poor compared to FOG INS,..." Do we know that? Please provide a reference to a source for that statement.

77: You have already defined GNSS

80: What other "flavors" are you referring to? 

85: What does loose, tight or ultra-tight mean? Please explain or provide a reference. 

86, 99. The sentence should not start with a reference. 

121. Questions 2 and 3 seem to be the same. And does question three refer to IHO S44 v6 order levels? If so, that should be made clear. 

133. Figure 1 is a map, so it should have a scale, north arrow, latitude and longitude reference graticules, and a legend. 

136. Low spectrum of what? Price? 

137. At the start of this paragraph "Table 1" is in bold, but not "Table 2".

142. Table 1: What is the GNSS model missing from the Applanix system? 

144. Table 3. Sometimes a star is used to indicate missing data, and sometimes a dash and sometimes a blank space. Be consistent.

146. With the INS installed on the roof, did you account for the offset to the vessel centre of rotation?

148. How do you know that that lever-arm offsets have less than 1cm of uncertainty? 

181. Please list the version number for each software package used 

182. You mention that ERS was used, and hence, the heave results are not discussed. Why are those two items related? Heave should not be ignored in ERS vertical referencing. More information is needed to make that statements. 

190. Figure 2. Was only PPK processing completed? If so, where was the base station located? It was not clear that all the GNSS data was post-processed with PPK from the methodology, 

197: suits should be suites 

260. Where does the uncertainty estimate come from? The boresite calibration result

263: Figure 4. It would be helpful to know the actual values of the patch test results and the uncertainty bars. 

290: It's not clear why this systematic bias would not be accounted for. From line 289 to 302, it's not clear how this discussion is relevant to the paper. What is the systematic bias you are examining, and why would it not be accounted for with the patch test / boresite calibration? More information is needed. 

304: Section 3.4. What ellipsoid is being used as the vertical reference for each system? Are you sure that they are all using the same coordinate reference system? If not, that might explain the vertical bias. 

323. Where is the "21" coming from? Again, it's not clear why this value is being included in the discussion. 

331. "We will discuss that in greater detail later". Where? in what section? 

333. Cycle slips related to the PPK processing? 

335. "reiterate". I don't see anywhere that you mentioned this before? 

377. Figure 8. The top three depth maps don't provide useful information. One depth map for the Wave-Master would be helpful, and then it would be very useful to see a difference map between the Ellipse-D and Wave-Master, and the FOG-3D and Wave-Master. That would clearly show differences in the final results and not rely on the uncertainty from the bottom half of the figure. It would also show if the uncertainty map is realistic. 

Overall the paper is very good, and I look forward to seeing it published; however, the items listed above need to be corrected before publication. 

Author Response

We do appreciate your constructive feedback and your time reviewing our manuscript. 

Please, see the attachment for our responses.

Reviewer 2 Report

The paper proposes the use of tactical grade GNSS - INS systems in shallow bathymetry modeling by taking a survey grade system as a reference. The authors performed fieldwork with simultaneous measurements of three systems and minimized the platform-based ambiguities or error sources by mounting them on a single platform. 

Although the study region is small, their findings are very interesting as they are asserting a very high degree of compatibility of tactical grade systems with respect to the survey grade system. 

1. As already mentioned in my original review, the paper proposes the use of tactical grade GNSS - INS systems in shallow bathymetry modeling by taking a survey grade system as a reference.  They are questioning if tactical grade systems can perform similar performance to the survey grade systems in shallow water bathimetry.

2. Topic is somehow original relevant to sensors topic due to above mentioned comparison  as they are comparing different grade GNSS sensors performance.

3. Performance of the tactical grade GNSS sensors in bathimetry seems new according to literature check of both Authors and mine. In addition, The authors performed fieldwork with simultaneous measurements of three systems and minimized the platform-based ambiguities or error sources by mounting them on a single platform.

4. Methodological steps are suitable in general. In accuracy assessment possible ambiguities due to the formal errors of the tactical system affecting the estimation of total uncertinities, which is already mentioned by the Authors as limitiations.

5. Yes the conclusions are consistent with findings and summarizes the answers to the main question (performance compariso of tactical grade GNSS), defining their limitations in accuracy assessment and advising the future research possibilities.

6. References are  less in number compared with other articles but they are mostly current and focused on the topic.

7. Fİgure 1 can be modified by adding a more genral map showing the geographic location of the study region. The quality of Figure 3-7 can be improved.

As I mentioned  the study region is small, additional field works can be performed (maybe in future studies).

Again as I mentioned there is a need  for the addition of  beam swath limitation findings in the Abstract (as it is in conclusions)

Author Response

(The authors gave the same response as above.)

Reviewer 3 Report

The topic of the submitted paper is interesting. The authors introduce a tactical-grade GNSS+INS for bathymetric survey. But this manuscript fails to prove the novelty of their method. Suggestions and comments are given as follows.

1.This manuscript is written as a technical report. The theoretical analysis is suggested to be added to prove the novelty of your method.

2. Though the detailed description of the experimental results of your test is interesting, some contents relate little to the novelty of your method.

3. It is stated “In this study, we address the application of affordable MEMS and FOG 47 INS, emphasizing their performances in multibeam sound and ranging (SONAR) survey procedures in shallow waters at sea, the limitations of the sensors, data acquisition, and processing challenges.”

The data acquisition and processing challenges are suggested to be clearly pointed out. And the novelty of your method should also be clearly pointed out.

4. It is stated “Therefore, we address the following questions: How close or far is the tactical-grade GNSS+INS positioning performance from the survey-grade systems in calm weather? Will the tactical-grade sensor meet shallow water bathymetric survey requirements?  For what survey order is it most suitable?”

These questions are answered using the results several tests. The theoretical analysis is lacked.

5. It is stated “Since all INS sensors are mounted on the same plate, we expect similar calibration offsets, assuming the variability between the INS body reference frames is negligible.”

The description of the test is not rigorous enough. Here, I only take it for example.

6. According to Table 3, the accuracy of INS determined by the gyros. Usually, the accuracy of INS is determined by the accelerometers. Can you give some explanations for your description? It is hard to learn the performance of the sensors according to Table 3.

7. A Table is suggested to be added to summarize your test results and prove the novelty of your method.

8. Overall, the manuscript is poorly organized and the novelty of the proposed method is not verified strictly.

Author Response

(The authors gave the same response as above.)

Round 2

Reviewer 3 Report

Compared with previous version, the english laguage and descirption of the paper are improved. However,most of the  suggestions and comments given by the reviewer are not handled carefully. And the reply to the suggestions of the reviwer is also lacked.

Author Response

Please, see our response in the attached Word document.
